# CalcAMP: A New Machine Learning Model for the Accurate Prediction of Antimicrobial Activity of Peptides

**DOI:** 10.3390/antibiotics12040725

**Published:** 2023-04-07

**Authors:** Colin Bournez, Martijn Riool, Leonie de Boer, Robert A. Cordfunke, Leonie de Best, Remko van Leeuwen, Jan Wouter Drijfhout, Sebastian A. J. Zaat, Gerard J. P. van Westen

**Affiliations:** 1Computational Drug Discovery, Drug Discovery and Safety, Leiden Academic Centre for Drug Research, Leiden University, P.O. Box 9502, 2300 RA Leiden, The Netherlands; 2Department of Medical Microbiology and Infection Prevention, Amsterdam Institute for Infection and Immunity, Amsterdam UMC, University of Amsterdam, 1105 AZ Amsterdam, The Netherlands; 3Department Immunology, Leiden University Medical Center, 2300 RC Leiden, The Netherlands; 4Madam Therapeutics B.V., Pivot Park Life Sciences Community, Kloosterstraat 9, 5349 AB Oss, The Netherlands

**Keywords:** antimicrobial peptides, artificial intelligence, bacteria, drug discovery, machine learning, antimicrobial resistance

## Abstract

To combat infection by microorganisms host organisms possess a primary arsenal via the innate immune system. Among them are defense peptides with the ability to target a wide range of pathogenic organisms, including bacteria, viruses, parasites, and fungi. Here, we present the development of a novel machine learning model capable of predicting the activity of antimicrobial peptides (AMPs), CalcAMP. AMPs, in particular short ones (<35 amino acids), can become an effective solution to face the multi-drug resistance issue arising worldwide. Whereas finding potent AMPs through classical wet-lab techniques is still a long and expensive process, a machine learning model can be useful to help researchers to rapidly identify whether peptides present potential or not. Our prediction model is based on a new data set constructed from the available public data on AMPs and experimental antimicrobial activities. CalcAMP can predict activity against both Gram-positive and Gram-negative bacteria. Different features either concerning general physicochemical properties or sequence composition have been assessed to retrieve higher prediction accuracy. CalcAMP can be used as an promising prediction asset to identify short AMPs among given peptide sequences.

## 1. Introduction

It is now recognized that an increase in bacterial resistance to conventional antibiotics can lead us to a “post antibiotic era” [1]. Conventional therapeutic strategies often no longer work; hence, there is an urgent need to find new drugs to fight pathogens. Despite a few promising compounds entering the different clinical phases, only two new classes (lipopeptides and oxazolidinones) were discovered in the last 20 years [2]. Moreover, both of them target only Gram-positive (Gram+) bacteria and their representatives already face serious resistance drawbacks [3,4]. Therefore, there is a clear priority to find new antimicrobial compounds, especially against a selection of critical strains published by the World Health Organization (WHO) [5]. Among alternatives to small molecules drugs, antimicrobial peptides (AMPs) are considered as interesting and promising candidates [6]. These peptides, which are already present in the innate immune system of plants, animals, and humans, possess both antimicrobial activity and immunomodulatory properties [7]. AMPs are an essential component of the body’s first line of defense against pathogens, even before the adaptive immune system is activated. Moreover, they exhibit diverse structural and functional profiles that can be optimized and fine-tuned to enhance their activity further [8,9]. As a result, they offer tremendous potential as novel therapeutic agents for combating a wide range of pathogens.

Several general properties are shared among AMPs such as a number of amino acids (AAs) between 5 and 60, typically a global net positive charge (>3), and amphipathic structures [10]. Still, even if the majority are cationic, several anionic AMPs exist [11,12]. Concerning their conformational characteristics, they show a great diversity of possible 3D structures from linear α-helices to β-sheets or random coils. They can also be cyclic or present with one or several disulfide bridges [10]. Figure 1 represents an overview of this variety in 3D structures among several AMPs.

In contrast to conventional antibiotics that inhibit specific intracellular targets, most AMPs act directly on the bacterial cell membrane or crucial cytoplasmic components [13]. Their interaction with the membrane provoke its disruption leading to the death of the bacteria [14]. Therefore, the threshold to develop resistance is higher since it involves a great modification of the membrane [15]. Moreover, since eukaryotic and prokaryotic membranes present different specifications, AMPs can be very selective against bacteria by accumulating at their negatively charged membrane surface [16]. In addition to their antibacterial effects, AMPs may also present antifungal, antiparasitic, antivirus, or even anticancer properties thus strengthening their potential and importance as new therapeutics [17,18,19,20]. However, despite their ability, numerous interesting peptides never passed preclinical stages for various reason. The most important one is their possible toxicity against human cells, in particular red blood cells, leading to their lysis similar to that of bacteria [21]. Peptides may also present some stability issues, low oral bioavailability, or high cost of production [22,23,24]. Those limitations are not unsurmountable with peptide modification and engineering, e.g., D-amino acids or N-terminal modification can be considered to improve their characteristics. Nevertheless, these challenges limited large pharmaceutical companies from fully supporting the development of AMPs [25]. As of today, only a few new AMPs are approved by the Food and Drug Administration (FDA) or are in clinical trials [26]. Therefore, in order to help in the discovery of new ones and reduce their cost, several computational approaches were developed.

In silico predictive models typically rely on the primary sequence of proven AMPs from which different compositional and physicochemical descriptors are calculated and used for predictions. Since the beginning of the century, several AMP prediction tools were conceived based on different features and various algorithms. Experimentally validated AMPs can be retrieved from different public databases such as DBAASP, DRAMP, or CAMP [27,28,29]. In general, such databases also include a predictive model on their website accessible via a webserver. In addition to these ones, several standalone tools have also been developed for this purpose. For instance, iAMP-2 utilizes a fuzzy k-nearest neighbor algorithm and pseudo amino acid composition (PseAAC) to predict not only antimicrobial activity but also other types of activities such as anticancer or antivirus activities [30]. Another tool, iAMPpred, employs a support vector machine (SVM) algorithm and three different types of features (compositional, structural, and physicochemical) [31]. More recently, Bahdra et al. presented AmPEP, which uses the distribution patterns of amino acid properties and a Random Forest (RF) classifier [32]. Subsequently, an updated version focused on shorter peptides (<30 AAs), named DEEP-AmPEP30, was introduced. DEEP-AmPEP30 is based on pseudo k-tuple reduced amino acids composition (PseKRAAC) and a convolutional neural network (CNN) algorithm [33]. However, the first deep learning-based AMP prediction model was conceived by Veltri et al. in 2018. It relies on the peptide’s primary sequence composition, converted to a numerous vector, for its prediction [34].

Still, the majority of current AMP prediction tools suffer from two main drawbacks. First, they do not account for differences in bacterial species or membrane structure differences, whereas the potency of AMPs can vary significantly depending on the target and the type of bacterial envelope [35]. Second, these tools employ randomly selected sequences without experimentally determined antimicrobial activity as the negative data set, rather than using confirmed inactive peptides. Nowadays, it remains difficult to develop a model specific to a bacterial species or a precise strain since little to no experimental data are available to do so. A few methods were conceived to more precisely target bacteria using their Gram classification and a threshold on activity to discriminate AMPs from other peptides (i.e., non-AMPs), but it is only specific to either Gram+ or Gram-negative (Gram−) bacteria [36,37]. Finally, even more recently, a new deep learning-based approach only specific to *Escherichia coli* has been published focusing on small AMPs (<20 AAs) without cysteine residues [38].

Here, we created a new data set composed exclusively of experimentally proven AMPs and Non-AMPs by setting an activity threshold to discriminate them. The experiments contained in our data set were focused on short AMPs (<35 AAs) since they can present potent activity coupled with low toxicity [15,39]. Furthermore, it is much more convenient to work with such peptides due to their small size, and they are simpler to synthesize, optimize and produce on a bigger scale, implying lower costs. Based on this data set, we introduced several novel predictive machine learning (ML) models separated according to the Gram classification. Hence, a specific prediction model was conceived for each class. In addition, an antifungal activity prediction model was created. The prediction method relies on the calculation of several sequence composition and physicochemical descriptors and several different ML algorithms assessed with cross validation (CV) and a holdout test data set.

## 2. Results

### 2.1. Exploration of the Data Set

#### 2.1.1. Global Overview

After classification of all bacterial species by their Gram staining (positive or negative), the exploration began with the search of the most tested species. Figure 2a shows a detailed overview of the five most retrieved species per category. For each category, the large difference observed between the first and second most tested microorganisms reveals a clear preference for certain species when running experimental tests. As suspected, *Staphylococcus aureus* and *E. coli* were the most tested Gram+ and Gram− bacterial species, respectively, and for fungi it was *Candida albicans*. While important species from the WHO priority list of antibiotic-resistant bacteria were present, namely *Acinetobacter baumannii* and *Pseudomonas aeruginosa*, other important species were not retrieved in the top five of tested microorganisms, such as *Helicobacter pylori*, *Neisseria gonorrhoeae*, or *Streptococcus pneumoniae* [2,5]. Figure 2b shows a Venn diagram illustrating the distribution of peptides tested against the different categories. As numerous peptides present several activities against different targets, their number is much lower than the number of activities. Moreover, the majority of them are common in both the Gram+ and Gram− data sets. However, there was also a significant number of peptides that are specific to each category (1243, 1600, and 576 for the Gram+, Gram−, and fungi, respectively). Concerning antifungal peptides, much less data are available than for bacteria and the majority of the peptides were retrieved within the Gram category. Still, a significant number of the antifungal peptides was also specific to this category. It appears that one global model for AMP prediction would not sufficiently represent the data since, as shown in Figure 2b, a great number of peptides were uniquely tested to one of the categories: Gram+, Gram−, or fungi.

#### 2.1.2. AMP/Non-AMP Peptides Analysis

The analysis of the overall amino acid composition (AAC) between AMPs and Non-AMPs is represented in Figure 3a. For both categories, the five most represented AAs were lysine (K), leucine (L), alanine (A), arginine (R), and glycine (G). In general, AMPs are known to be cationic, and they indeed exhibited a higher frequency of positive residues such as lysine or arginine. Additionally, tryptophan (W) was overrepresented in AMPs. Conversely, the proportion of histidine (H) was slightly higher for Non-AMP peptides, which were also much more enriched in negative residues such as aspartic acid (D) and glutamic acid (E), despite these being in low abundance overall. Finally, a global similarity was observed for non-polar residues such as phenylalanine (F), isoleucine (I), leucine (L), and cysteine (C) participating in the amphiphilic properties of these peptides. These results were in correlation with the global charge difference between the two categories (Figure 3b). As mentioned above, AMPs were significantly more positively charged than Non-AMPs (Mann–Whitney U test, *p* < 0.05). They had an average positive charge of +4.8 (standard deviation SD: 2.70) vs. a charge of +2.8 (SD: 2.56) for Non-AMPs. AMPs were also significantly heavier (Mann–Whitney U test, *p* < 0.05) with a mean of 2241 g·mol^−1^ (SD: 703) vs. 1911 g·mol^−1^ (SD: 715) for Non-AMPs (see Figure 3c). Both a PCA and t-SNE analysis were performed on the overall physicochemical descriptors (Appendix A) and AAC (Appendix A). Such projections allow us to quickly see if one can perceive a separation between AMPs and Non-AMPs. A significant overlap existed between the two categories in the PCA space. For t-SNE, the projections were very sparse and small clusters appeared to be quite discriminative but overlap was visible. Therefore, we hypothesize that the specific AA sequence is more important for the biological activity than the overall composition.

A deeper analysis on the 4174 common peptides tested against both Gram+ and Gram− bacteria and the distribution of AMPs/Non-AMPs is presented within the matrix in Figure 4 (see *Data labelling* section for details on the classification). The majority, 89%, acted either as an AMP or as a Non-AMP in both Gram+ and Gram− categories. Still, a non-negligible part of Gram− bacteria AMPs (7%) were not active against Gram+ bacteria and vice versa (4%). These results reinforce the importance to have a specific model for Gram+ and Gram− bacteria rather than a global one.

A PCA and a t-SNE projection were applied based on several general physicochemical descriptors and were subsequently colored by class in Figure 4 matrix. Based on the PCA projection, Figure 5a, a slight separation appeared between AMPs (right) from Non-AMPs (left). However, a significant overlap remained between them. Peptides labelled as AMP for Gram+ and Non-AMP for Gram− (in green) tended to cluster with the common Non-AMP ones on the left side. Whereas peptides labeled AMP for Gram− and Non-AMP for Gram+ (in black) tended to be projected with the common AMPs on the right side. The correlation circle associated with this PCA (Appendix A) shows that component 1 is mostly “Charge” and component 2 is “MW” (molecular weight) and “Length”. The plot confirms that AMPs have on average a higher molecular weight and are more positively charged. The same separation was retrieved in the t-SNE projection, Figure 5b, with an interesting cluster on the bottom of practically only the Non-AMPs projected there. Still, in most cases, AMPs and Non-AMPs are near coincident in the plots, meaning they present similar physicochemical characteristics. The same projections were also produced using AAC (Appendix A). These projections show an horizontal separation between AMPs and Non-AMPs with AMPs being much more dispersed than Non-AMPs. The separation was realized for arginine (R), lysine (K), tryptophan (W), and leucine (L) for the top part (AMPs) and the other AAs for the bottom part (Non-AMPs).

### 2.2. Antimicrobial Activity Prediction

#### 2.2.1. Feature Selection

As a basis for the prediction model, it is crucial to extract and select features from the peptide’s primary sequence. The features from the sequences can be divided into two main categories: based on physicochemical descriptors or based on AAC. For this study, both types were used and evaluated individually to retain only the most interesting ones. Feature selection was performed using the Random Forest classifier (RF) [40], a ML algorithm that has an extensive track record in both drug discovery and AMP prediction and can be interpreted. Hence, this algorithm was selected in order to obtain features that produce the most accurate peptide classification. This first preliminary assessment, based on a classical RF-classifier, was performed with 10-fold cross validation (CV) experiments each time. The CV process consists of the data set being split into k-folds and k − 1 folds being used as training data while the final fold is retained for evaluation. Therefore, the model is assessed k times where each of the k-folds serve once as the validation data. After that, the mean score and standard deviation for each metric can be calculated. 

For sequence-based features, the AAC, the dipeptide composition (DPC), the pseudo amino acid composition (PseAAC) [41], and the composition-transition-distribution (CTD) descriptors [42] were retained. AAC is the frequency of each AA in the sequence and DPC is the same for dipeptides. Thus, they are made of 20 and 400 descriptors, respectively. Contrarily to those two, PseAAC allows us to retain all the sequence-order information. Depending on the initial parameters, it contains a different number of descriptors, derived from the primary sequence and incorporating some sequence-order knowledge. Finally, CTD descriptors are divided in three components: composition (C), the number of AAs with a particular property divided by the length of the sequence; transition (T), the frequency where AAs with a particular property is followed by AAs of another property; and distribution (D), the measure of different lengths of the sequence of the distribution of each property. In total, CTD descriptors are composed of 147 features (21 for C and T and 105 for D) and describes seven properties: charge, hydrophobicity, normalized van der Waals volume, polarity, polarizability, secondary structure, and solvent accessibility. Concerning the global physicochemical (GPC) descriptors, they are composed of the 10 following descriptors: length, MW, global charge, charge density, isoelectric point, instability index, aromaticity, aliphatic index, Boman index, and hydrophobic ratio. Of note, AA scale-based descriptors such as T-scales or Z-scales as well as descriptors derived from the AA index were dropped because of insufficient prediction results even though they were previously shown to perform well on peptide bioactivity modelling [43,44]. 

The performances were evaluated according to several metrics including accuracy, sensitivity, specificity, area under the receiver operator characteristic (ROC) curve (AUC), and Matthew’s correlation coefficient (MCC). The results are shown in Table 1. For all sets of features, individually and aggregated, the predictions between categories were similar and no major difference appeared. For both categories, CTD descriptors were the ones with the best prediction according to all metrics tested. It makes sense that CTD are the descriptors producing the best predictions since they are the most complete ones, including sequence composition and physicochemical criteria. The higher scores in all metrics was obtained using all these set of descriptors together. The choice of these features allowed our basic RF model to achieve an accuracy of 81% in both categories.

#### 2.2.2. Algorithm Choice

To assess which algorithm best suits our prediction purpose, several different models from classical classification algorithms were tested. Thus, the performance of 14 ML algorithms and one Multi-Layer Perceptron (MLP) was evaluated with 10-fold CV each time (Appendix A). For both categories, i.e., Gram+ and Gram−, it was clearly observed that ensemble tree-based algorithms outperformed, in all metrics, all the other types of ML models as well as the MLP. They achieved their prediction with a mean accuracy of 80% compared to 73% for k-nearest neighbors, for example. Therefore, the decision was made to build and tune a model for each top five tested algorithms in each category: Catboost [45], LightGBM [46], XGBoost [47], Random Forest, and Extra Trees [48].

#### 2.2.3. Performance and Interpretation

For each algorithm selected, two models were created and tuned, one with all the features (601) and one with a supplementary feature selection process. Indeed, feature selection can be an important step to accelerate the learning and training but also improve the performance of any model [49]. In order to identify the best discriminative and useful features, their importance weights were used followed by a recursive elimination with 3-fold CV each time, reducing their numbers to 75. The overall results of each AMP prediction model is presented in Appendix A. For both categories, no created model stood out, as they all presented similar results within their standard deviation ranges. Moreover, no significant performance losses were observed after our feature selection step. The selection of the best model was therefore made using the external data set. For the Gram+ model (CalcAMP+), the best classifier was the Extra Trees one with all features, as it achieved a prediction accuracy on the external test set of 79% and an MCC of 0.58. For the Gram− model (CalcAMP-), the best one was obtained with the LightGBM algorithm using all of the features. This model obtained an accuracy of 80% and an MCC of 0.61. More results can be found in Table 2 in the *Comparison with Other Prediction Tools* Section 2.2.4.

A “SHapley Additive exPlanations” (SHAP) values [50] analysis was performed to globally interpret the predictions on our test data set. SHAP values allow us to visualize which features are important for the prediction and their contribution. For the two models, the top 20 variable importance plot is shown in Figure 6, with their impacts on the prediction. The features are ranked in descending order and the horizontal scatterplot for each illustrates whether the effect of that feature was associated with a positive or a negative prediction output. For example, for CalcAMP+, a high MW (dots in red) had a strong and positive impact on AMP prediction. Of these top 20 features, only three were common to both models: MW, Charge, and pI. The majority of the top features were from CTD descriptors. They are identifiable by their names beginning with an underscore character, followed by the property and finally the component characteristics: composition (C), transition (T), and distribution (D). From the AAC, only the proportion of tryptophan (W) was represented in CalcAMP+. No features from DPC descriptors and only one from PseAAC were retrieved in each model. For the global physicochemical descriptors, four out of ten were part of the top features in both models, meaning that their importance was high, as those of the CTD descriptors. For CalcAMP+ (A), the impact of the 20 features were quite important as illustrated on the horizontal distribution, whereas for the CalcAMP- (B), except for charge and MW, the impact of the other features were less important.

A deeper look into the prediction results, especially at the confusion matrix associated with the predictions (Figure 7a), reveals that the great majority of true positives (TP) and true negatives (TN) was similarly predicted in both models. However, the image was less clear for false positives (FP) or false negatives (FN), where a little more than half were predicted differently even if the overall metrics of both models were equivalent. This confusion matrix shows that our models returned different prediction results and the distinction between Gram+ and Gram− bacteria remains important. Finally, the analysis of the probabilities associated with the prediction and not directly the binary output (Figure 7b) displayed a scoring difference between our two models. CalcAMP+ returned lower scores in general for any category, while CalcAMP- had an average close to 1 or 0 for TP and TN predictions. However, the important observation here is that there was a significant scoring difference between TP and FP and between TN and FN for both models. TP scores were significantly higher than FP scores (Mann–Whitney U test *p* < 0.05) and TN scores were significantly lower than FN scores (Mann–Whitney U test *p* < 0.05). Therefore, in order to increase the sensitivity or the specificity of the prediction, one should increase or decrease the classification threshold differently for the CalcAMP+ and CalcAMP- models since they had different scoring scales.

#### 2.2.4. Comparison with Other Prediction Tools

Previously published AMP prediction tools were assessed in order to compare them to our models using the external benchmark data set, composed of AMPs and non-AMPs common to both Gram+ and Gram− bacteria. As already mentioned, most AMP predictive approaches that have been developed are also based on a training set composed of AMPs found in public databases. However, their negative data set is made without an activity threshold but based on random peptide sequences from the UniProt database tagged as Non-AMP. In our case, the method is significantly different since the classification of peptides as AMP or Non-AMP was based on their measured activities (see *Data Labelling*, Section 4.1.2). Nevertheless, our best classifiers were compared to five existing prediction models: iAMPpred, DBAASP, RF-AmPEP30, Deep-AmPEP30, and AMP Scanner Vr.2. These five tools are freely available as webservers, which were used for this comparison. 

The comparison demonstrates the superiority of the CalcAMP models over all other tested tools in the global prediction of activity (Table 2). The CalcAMP accuracy was 79% and 80% for Gram+ and Gram−, respectively, versus 67% or less for the others, and CalcAMP demonstrated an MCC of at least 0.58 versus 0.35 or less. However, except for the DBAASP model, all the others had a higher sensitivity (>90%), meaning that they were more prone to predict the peptides as an AMP than CalcAMP. These results can be explained by their different training data sets. For models using randomized negatives, any peptide with a recorded antimicrobial activity is an AMP so in our external data set most of them will be predicted as an AMP. However, the drawback is their lower specificity (≤30%), implying a difficulty in discriminating and predicting peptides as Non-AMP if they have a weak activity on their target. Both CalcAMP models presented a balanced high specificity and sensitivity. They were able to efficiently differentiate between peptides with high activity and those with lower ones. Figure 8 shows the different ROC curves (except for DBAASP model) and confirm that our models have high accuracy at various thresholds and are superior to the other models.

**Table 2 antibiotics-12-00725-t002:** Comparison of different AMP prediction classifiers using the external data set. Bold values indicate the best value per column.

Model	Accuracy	Sensitivity	Specificity	AUC-ROC	MCC
Deep-AmPEP30 ^1^	0.60	0.92	0.29	0.71	0.27
RF-AmPEP30 ^1^	0.59	**0.94**	0.25	0.74	0.26
AMP_Scanner	0.61	0.93	0.30	0.75	0.29
iAMPpred	0.60	0.91	0.29	0.67	0.26
DBAASP	0.67	0.74	0.61	- ^2^	0.35
*Average*	*0.61*	*0.89*	*0.35*	*0.72*	*0.29*
CalcAMP+	0.79	0.79	0.79	0.86	0.58
CalcAMP-	**0.80**	0.78	**0.82**	**0.87**	**0.61**

^1^ For Deep-AmPEP30 and RF-AmPEP30 models, only peptides with a length between 5 and 30 AAs were used since it does not predict using longer ones. ^2^ For DBAASP, AUC-ROC cannot be calculated and the ROC curve could not be displayed since it only returns binary results and the probabilities associated are not accessible.

To further compare CalcAMP with the other tools, an assessment on their own respective external data set was also performed using the external benchmark for AmPEP and a shortened version of the Antimicrobial Peptide Scanner vr.2 validation data set (only peptides with a length between 5 and 30 AAs). More details and results can be found in Appendix A, section *Comparison with other datasets*. As expected, CalcAMP did not perform as well on those data sets as on our own external data set but still displayed an accuracy of around 70% and an MCC > 0.4. Such a loss of performance can once again be explained by the initial labelling difference. Indeed, most peptides labelled as AMPs in those data sets would be labelled Non-AMP in our case, thus accounting for this decrease in prediction power. However, in doing these extra comparisons, our main interest was the Non-AMPs predictions and therefore the sensitivity. Indeed, most or all of their Non-AMPs have never been seen by any of our models. For both external data sets, the sensitivities of CalcAMP+ and CalcAMP- were higher than all the other models, reaching 100% for CalcAMP+ on the Antimicrobial Peptide Scanner vr.2 validation data set (Appendix A).

### 2.3. Antifungal Activity Prediction

Fungal infections still remain a serious threat for humans and, similar to antibiotics, antifungal drugs present some limitations [51,52]. Therefore, antifungal peptides (AFPs) have emerged as new potential treatments to prevent or treat such infections [53], similar to AMPs for bacteria. Even though the initial and main focus here was AMPs, quite a few reported antifungal activities caught our attention. However, given the inferiority of input data (1301 Non-AFPs and 887 AFPs), a lighter protocol was proposed and only models based on RF and ET algorithms were employed. Concerning the initial exploration of the data set with global physicochemical descriptors, no clear separation between AFPs and Non-AFPs was visible with either PCA or t-SNE projections. Similarly, when looking at AAC, no AAs were unbalanced between the categories, with two exceptions. Arginine was enriched in AFPs over Non-AFPs and for alanine the inverse was true. Additionally, like with AMPs, the five most retrieved amino acids were lysine, leucine, glycine, alanine, and arginine. Of note, this could be the result of a bias as these data were retrieved from AMP public databases. Therefore, most of the peptides were synthetized with the aim to target bacteria so they present the characteristics of AMPs.

#### 2.3.1. Performance and Interpretation

Following the same protocol as for the AMPs, two models per algorithm were created and tuned: one with all of the features (601) and one with a feature selection step ending with 75 features. Again, no created model stood out as they all presented similar results within their standard deviation ranges (Appendix A). No significant performance losses were observed after our feature selection step either. On the external test set consisting of 30 AFPs and 30 Non-AFPs, the best classifier was the Random Forest one with all features (CalcAFP) with an accuracy of 77% and a great specificity of 90% but a lower sensitivity (63%). The prediction results are presented in Table 3. Such a difference might be explained by the small input imbalance, where roughly 60% were Non-AFPs versus 40% AFPs (due to the small amount of data, the choice was made to leave it as is). Therefore, in our model it is best to discard Non-AFPs from selection rather than identifying the AFPs. 

Analysis of the model with SHAP values and the top 20 variable importance plot are shown in Figure 9. Similar to the AMP prediction, charge and MW were important features; the higher they were, the higher the positive impact on AFP prediction. However, the descriptor pI was not retrieved in the top 20 features. In correlation with the AAC comparison, the proportion of arginine (R) and also the dipeptide RR were retrieved, and both were highly correlated with AFP prediction. Finally, the presence of three PseAAC descriptors showed that they might be more important and interesting for AFP prediction than for AMP predictions.

#### 2.3.2. Comparison with Other Prediction Tools

The comparison results with other AFP prediction tools is provided in Table 3. Three available tools were evaluated: iAMPpred, ClassAMP [54], and AntiFP [55]. To note, iAMPpred and ClassAMP are not specific AFP prediction tools but multilabel ones proposing AFP prediction. For ClassAMP, the model based on the SVM algorithm was selected. Sequences predicted as antifungal were labeled as AFP while the ones not predicted as antifungal but as other classes were considered Non-AFPs. Similar to the previously developed AMP prediction tools, these tools were developed based on a training set composed of AFPs from public databases without an activity threshold and random peptide sequences from UniProt or Swiss-Prot tagged as non-AFP. These three tools are freely available as a webserver, which we used for this comparison. CalcAFP achieved an accuracy of 77% in contrast with 48% to 52% for the other predictors. It clearly outperformed them on all metrics, except sensitivity where iAMPpred was higher (63% vs. 77%).

Another data set was also evaluated, the one from the Antifp tool. The Antifp_Main validation data set was modified to keep only peptides with a length between 5 and 35 AAs. The details and results can be found in Appendix A, section *Comparison with other datasets*. Unfortunately, with this data set, CalcAFP performance suffered heavy losses in all metrics with an MCC of −0.12 and was not able to perform better than a random model. Still, it maintained a good specificity of 79% (Appendix A). Hence, the performance difference can be explained by the different methodologies and initial classification of AFPs/Non-AFPs (Appendix A). Even though our results for AFP prediction were lower than for the AMPs predictions, to the best of our knowledge, CalcAFP is the only model that works exclusively with peptides having experimentally measured antifungal activity and classified AFP/Non-AFP using a threshold.

## 3. Discussion

The increase in antibiotic resistance urges the discovery of new therapeutics to tackle this issue. AMPs represent a very interesting alternative to small molecules against bacteria. They can have a large spectrum of activity against bacteria (either Gram+, Gram−, or both classes) and fungi but also against viruses, parasites, or even cancer cells. Unlike eukaryotic cells, prokaryotic cells, and particularly their cytoplasmic membranes, are negatively charged and thus it is more convenient for the action of small cationic peptides. Therefore, AMPs tend to be more enriched in arginine or lysine residues compared to peptides showing no antimicrobial activity. However, an overall positive charge alone and the proportion of certain AAs are not enough to correctly discriminate AMPs from Non-AMPs. Other descriptors are important and the combination of several physicochemical and compositional ones is the key to build an efficient prediction model. Here, the creation of a novel public data set allowed us to construct ML models for antimicrobial activity prediction. For each class of bacteria, Gram+ or Gram−, a general accuracy of around 80% was achieved. CalcAMP outperformed existing AMP classifiers and was also able to correctly classify Non-AMP data set from these other tools.

The main limitation of the models comes from the input data gathered from different sources that present quite high heterogeneity. Moreover, it can be difficult to classify a peptide as a generic AMP since there is no clear experimental test to define it. Indeed, several factors can influence the outcome, such as the bacterial strain tested, the growth medium, and the type of activity measured. Since prediction models rely on input data, small changes in the method and choices to discriminate AMPs from Non-AMPs can have great consequences on the output. In an ideal world, models would be species or even strain specific, but as discussed in the introduction a lack of data makes this quite difficult to achieve. Future work will also focus on toxicity prediction with the creation of a new model coupled to the current one allowing one to have both activity and toxicity predictions returned. Indeed, toxicity remains a major issue in AMP design and development; therefore, an effective toxicity prediction tool would be a significant help for the design of potential clinical AMPs. We hope our method can be of great help and thus accelerate the R&D process of finding new AMPs as a potential alternative to antibiotics. Moreover, we have made our current curated data set available for use which could serve as a basis for other experiments and development of tools.

## 4. Materials and Methods

### 4.1. Data Preparation

#### 4.1.1. Data Mining and Preprocessing

The data set of peptides serving as input for the different prediction models was built with publicly available data from different databases. At first, seven databases were selected and manually mined: ADAM [56], BaAMPs [57], CAMP [29], DBAASP [27], DRAMP [28], LAMP2 [58], and YADAMP [59]. From these, ADAM, having no precise experimental data on activity, and BaAMPs, containing no relevant data (only activities on biofilms), were rejected. Table 4 lists the databases examined with their URL and number of corresponding peptides.

A primary filter was applied to the selected databases to keep only activity against either bacteria or fungi; all other activities (virus, parasites, cells, etc.) were discarded. Then, a second filter was implemented to retain only interesting activity types such as minimal inhibitory or minimal bactericidal concentrations (MIC/MBC) or 99.9% lethal concentration (LC_99.9_). Therefore, nonstandard or unclear activity types were rejected. In order to easily and accurately compare experimental activity values, they were all converted to µM, the most predominantly used, using Equation (1). Finally, since the study focuses on the prediction of natural short AMPs, only peptides having a length between 5 and 35 AAs were considered. Among them, repetitive sequences of single AAs (e.g., RRRRRRR, AAAAAAAAA) and sequences containing unnatural AAs were left out. The full data set is available in the Appendix A.
(1)$$\left(µM\right)=\frac{C(µg\cdot mL^{-1})}{MW(Da)}\times 1000,$$

#### 4.1.2. Data Labelling

Each activity recorded was grouped by Gram classification using the bacteria species tested. Unlike existing tools, with a negative data set constructed by selecting random sequences from UniProt, [32] our strategy was to work exclusively with peptides that were experimentally tested. Depending on their activities, peptides were classified either as AMP (1) or Non-AMP (0) based on reported MIC/MBC or equivalent LC_99.9_ values for antimicrobially active AMPs [60]. The activity threshold was set at ≤15 µM for a peptide to be considered as active (strong) and above 25 µM for inactive ones (or weaker AMPs that need higher concentrations to show antimicrobial properties). The values between these thresholds were discarded as they are considered in an area where labelling was not certain enough. The majority of peptides have been tested against multiple species of bacteria and hence present several activity values. Therefore, to take into account this heterogeneity, a specific workflow was set up for each peptide. If all activity values belonged to the active category (Gram+, Gram−, or fungi; ≤15 µM) or if the majority was in this category while none of the values are above 25 µM, then the peptide was labelled as an AMP (1). Conversely, if all activity values are higher than 25 µM or if the majority is and none are below 15 µM, then the peptide was labelled as Non-AMP (0). In every other case, the antimicrobial activity was considered as unsure, and the peptide could not be labelled as either active or inactive and was discarded for the creation of the prediction models. This method is a means for us to tackle the heterogeneity of the experimental data and to take into consideration the uncertainty of antimicrobial activity caused by different experimental conditions and/or experimental errors. In Figure 10, different examples of labelling depending on the situation of the peptide and its related activities are displayed. The categories were balanced between AMPs and Non-AMPs; there were slightly fewer AMPs for the Fungi category but no major imbalance was present. Our final data set is composed as follows:Gram+: 5791 peptides; 2849 Non-AMP (49%) and 2942 AMP (51%)Gram−: 6087 peptides; 3163 Non-AMP (52%) and 2924 AMP (48%)Fungi: 2544 peptides; 1475 Non-AMP (58%) and 1069 AMP (42%)

**Figure 10 antibiotics-12-00725-f010:**
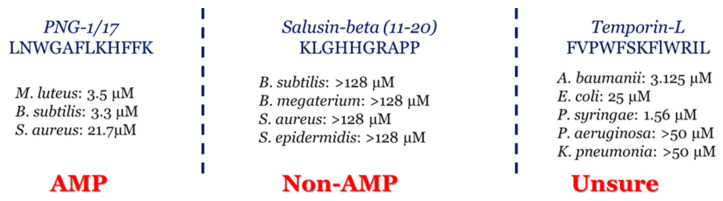
Different examples of peptides with their reported experimental activities and their label.

#### 4.1.3. Creation of the Data Sets

To build training data sets and in order to avoid redundancy, the software CD-HIT [61] was used with a cutoff of 95% to remove highly similar sequences for both AMPs and Non-AMPs distinctly before aggregating them. For the Gram− category, since there was a small imbalance between AMPs and Non-AMPs, 350 Non-AMPs sequences were removed randomly before the model construction. The external data set test was composed of 350 peptides (175 AMPs and 175 Non-AMPs). It was used for the evaluation and comparison of our models to the previously developed models. For its creation, all the common AMPs and Non-AMPs of the Gram+ and Gram− data set were assembled. Then, a clustering was performed to select 300 representative peptides of each category. It was followed by filtering using CD-HIT with a cutoff of 80%. Finally, this was followed with a random selection on the remaining peptides to reach 175 AMPs and 175 Non-AMPs. Figure 11 shows how the training set was representative of the entire data set via a PCA projection based on physicochemical descriptors.

Concerning the AFP prediction external data set, a simple clustering on our training data set to select 30 AFPs and 30 Non-AFPs representatives was performed.

### 4.2. Machine Learning Experiments

#### 4.2.1. Feature Calculation

A preliminary exploration and assessment of the different existing types of descriptors (single, double, tri-peptide composition, Moran, Geary, Moreau-Broto, etc.) and co-variance-encoding methods (auto, cross, and auto-cross using different AA descriptor scales) was achieved. The ones that showed insufficient performance with a simple RF model were discarded and only the most promising were kept for the rest of the study. Thus, only AAC, DPC, CTD, GPC, and PseAAC (see section “Feature selection” for their brief description) were retained. All the calculations were made using Python 3.7 and the packages modlAMP 4.3.0 and PyBioMed 1.0.

#### 4.2.2. Model Comparison

The comparison of different ML algorithms to develop a classification model between AMP and non-AMP were made using the package PyCaret 2.3.6. PyCaret is an open-source, low-code library allowing users to quickly compare several different ML algorithms. The algorithms included were Random Forest (RF), Extra Trees (ET), Extreme Gradient Boosting (xgboost), Light Gradient Boosting Machine (lightgbm), Ada Boost (ada), Gradient Boosting (gbc), CatBoost (catboost), Logistic Regression (lr), SVM linear kernel (svm), Naive Bayes (nb), Decision Tree (DT), Ridge (ridge), K-Nearest Neighbor (knn), Quadratic Discriminant Analysis (qda), Linear Discriminant Analysis (lda), and a Dummy Classifier (dummy). In addition, a Multi-layer Perceptron model created with Scikit-Learn 0.23.2 was added for the comparison step. The comparison was made using the “Classification” modules, without changing the parameters, and for each category (Gram+, Gram−), the top five models were kept for further analysis.

#### 4.2.3. Model Creation and Tuning

For each retained algorithm, two models were created and tuned, one with all the features and one with a set of 75 features selected. The packages lightgbm 3.1.1, xgboost 1.5.0, and catboost 0.26.20 were used for the creation of our LightGBM, XGBoost, and CatBoost classifiers, respectively. For our RF and ET classifiers we used Scikit-Learn 0.23.2. The tuning and optimization of our created models were performed with the Scikit-Learn API and the “RandomizedSearchCV” function (3-fold CV per change of parameter) on the whole training data set. Once the hyperparameters were selected, the final model was established with them and the performance was evaluated first via a 10-fold CV and then with the external test set.

#### 4.2.4. Feature Selection

Once the model calculated with all features was optimized, the process of feature selection started with the help of feature importance weights. First, Scikit-Learn API with the “SelectFromModel” function was used to remove unimportant features. Then, the recursive feature elimination (RFE) module was used, with 3-fold CV, to select 75 features from those remaining. RFE selects best features by recursively removing the least important features until the desired number of features is reached. The choice of 75 features was determined by plotting accuracy vs. number of features and the appearance of a plateau from this number.

#### 4.2.5. Evaluation Metrics

The developed classifiers were systematically assessed using a 10-fold CV and five metrics: Accuracy (Acc), Sensitivity (Sen), Specificity (Spe), area under the ROC curve (AUC), and Matthew’s correlation coefficient (MCC). The external test set, with data unseen by our models, was not included into the CV process and serves for supplementary evaluation and comparison. The metrics are defined as follows:(2)$$Acc=\frac{TP+TN}{TP+FP+TN+FP},$$
(3)$$Sen=\frac{TP}{TP+FN},$$
(4)$$Spe=\frac{TN}{TN+FP},$$
(5)$$MCC=\frac{\left(TP\times TN)-(FP\times FN\right)}{\sqrt{\left(TP+FP\right)\times \left(TP+FN\right)\times \left(TN+FP\right)\times \left(TN+FN\right)}},$$
where TP (True positive) is the number of correctly predicted AMPs, TN (True Negative) is the number of correctly predicted Non-AMPs, FP (False Positive) is the number of Non-AMPs incorrectly predicted as AMPs, and FN (False Negative) is the number of AMPs incorrectly predicted as Non-AMPs. AUC is the area under the ROC curve: the plot of the true positive rate (sensitivity) as a function of the false positive rate (1—specificity). Accuracy (Equation (2)) is a global metric representing the sum of true positives and true negatives divided by the total number of the data; it indicates the proportion of correct predictions. Sensitivity (Equation (3)) and specificity (Equation (4)) focus on how well the classifier predicts AMPs and non-AMPs, respectively. AUC measures the ability to correctly distinguish between classes. All of them are between 0 and 1, and the higher value the better the performance of the model. Matthew’s correlation coefficient (Equation (5)) also measures the overall quality of a binary classifier and is widely used in the field of ML. The MCC value is between −1 and 1. Again, the closer to 1, the better the performance, and a value of 0 indicates that the model is no better than a random prediction and −1 is a total disagreement between prediction and reality.

### 4.3. Implementation

All ML experiments were performed and implemented using Python 3.7. All the figures are made using Matplotlib and Seaborn packages. Input, generated, or analyzed data used in this study are included in this article’s Appendix A sets or uploaded in Zenodo: https://doi.org/10.5281/zenodo.7588702. The code is available at: https://github.com/CDDLeiden/CalcAMP.

## Figures and Tables

**Figure 1 antibiotics-12-00725-f001:**
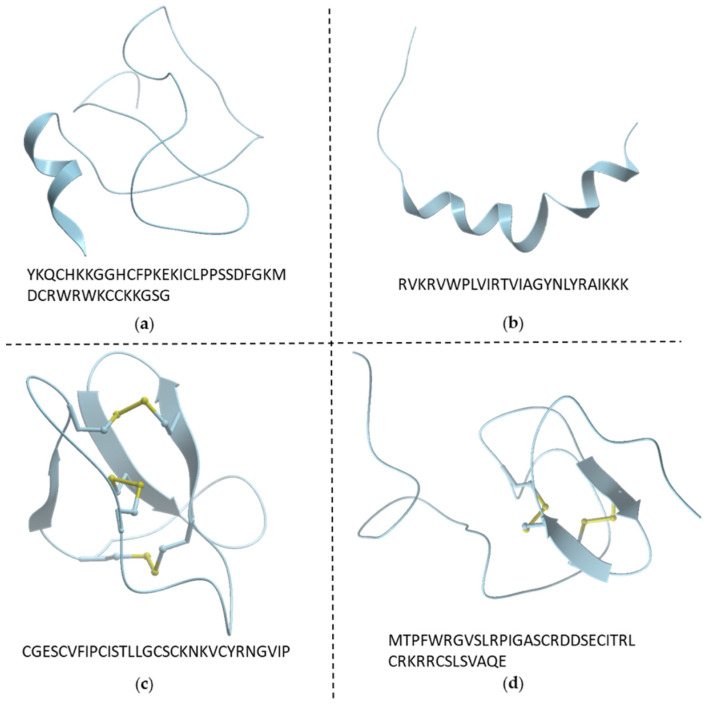
Overview of the variety of 3D AMP structures. (**a**) Crotamine from PDB code 1Z99; (**b**) fowlcidin from PDB code 2AMN; (**c**) circullin B from PDB code 2ERI; (**d**) LEAP-2 from PDB code 2L1Q. Yellow bonds represent disulfide bridges.

**Figure 2 antibiotics-12-00725-f002:**
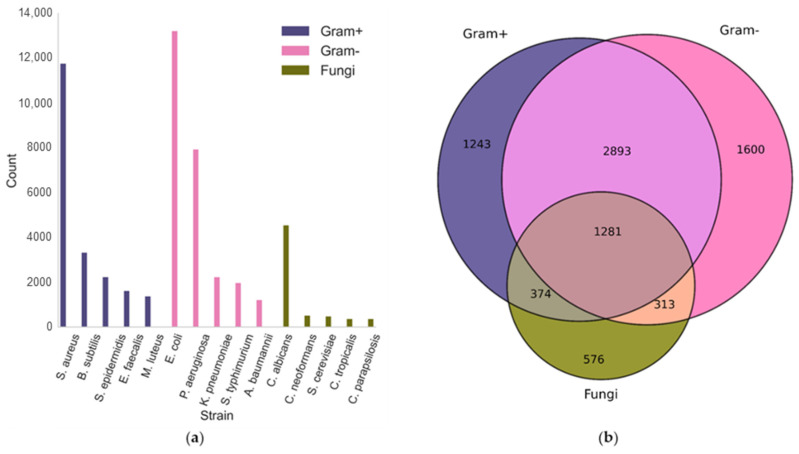
Number of experimental assays retrieved for the top five species by category (**a**). Venn diagram showing the distribution of peptides per category (**b**).

**Figure 3 antibiotics-12-00725-f003:**
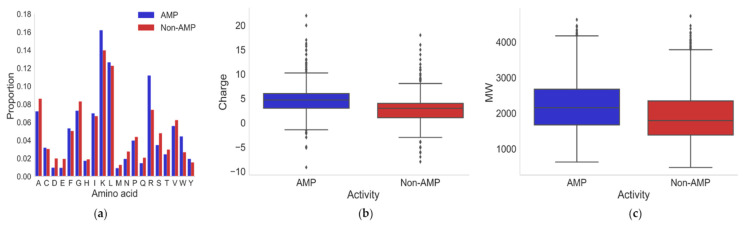
Comparison of amino acid composition (**a**), global net charge (**b**), and molecular weight (**c**) between AMPs and Non-AMPs.

**Figure 4 antibiotics-12-00725-f004:**
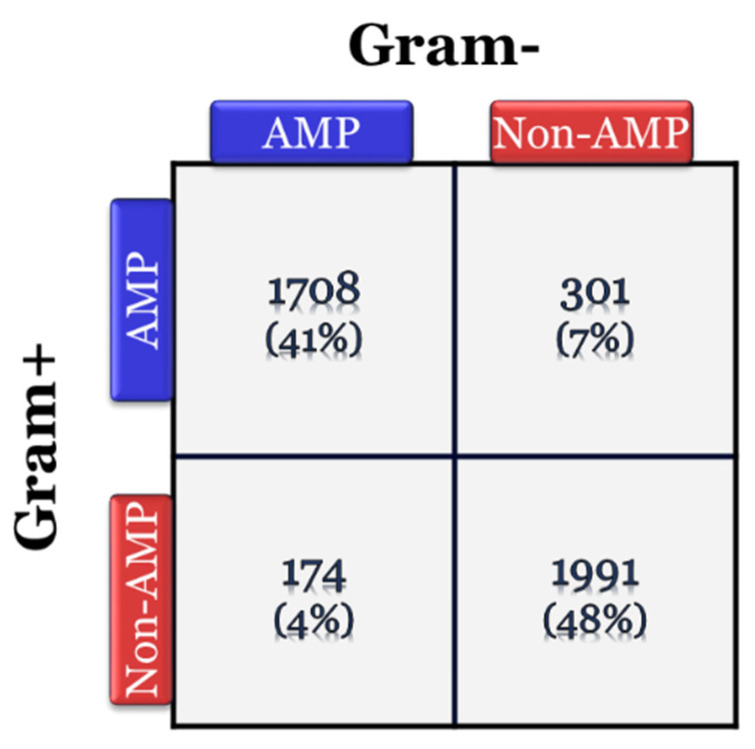
Matrix of labels for the common peptides between Gram− and Gram+ categories.

**Figure 5 antibiotics-12-00725-f005:**
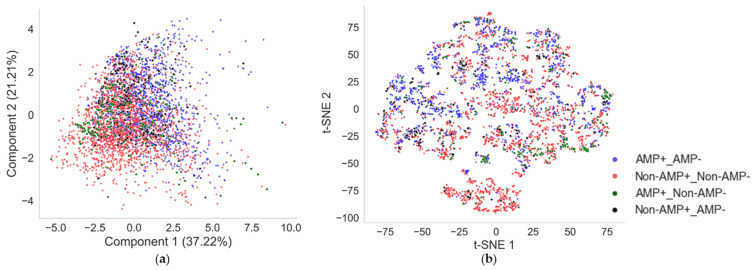
PCA (**a**) and t-SNE (**b**) projections of physicochemical descriptors between common peptides of Gram+ and Gram− categories. In blue are peptides are labelled as AMP in both categories, in red are peptides are labelled as Non-AMP in both categories, in green are peptides labelled AMP for Gram+ and Non-AMP for Gram−, and in black is the opposite.

**Figure 6 antibiotics-12-00725-f006:**
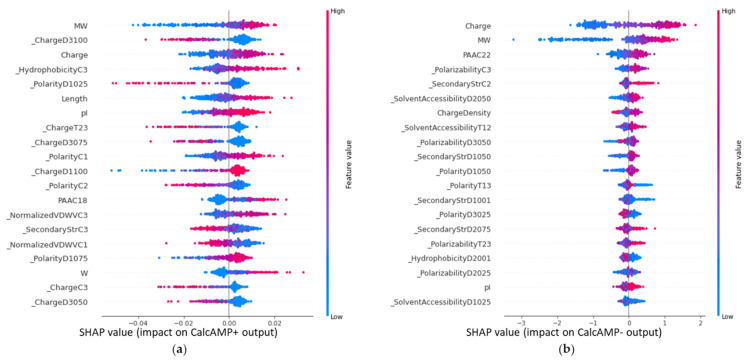
Top 20 features importance plot and their impact on the external test set prediction for CalcAMP+ (**a**) and CalcAMP- (**b**). Shown are physicochemical properties such as molecular weight (MW), Charge, or Length. However, the majority of the top features were from CTD descriptors. They are identifiable by their names beginning with an underscore character, followed by the property and finally the component characteristics: composition (C), transition (T), and distribution (D).

**Figure 7 antibiotics-12-00725-f007:**
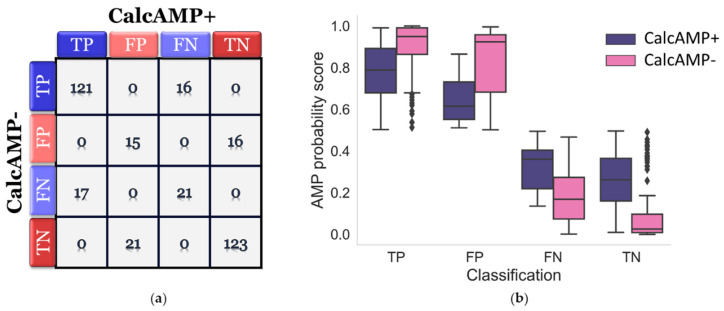
Confusion matrix for CalcAMP+ model prediction versus CalcAMP- (**a**) and AMP probability score by predicted class (**b**).

**Figure 8 antibiotics-12-00725-f008:**
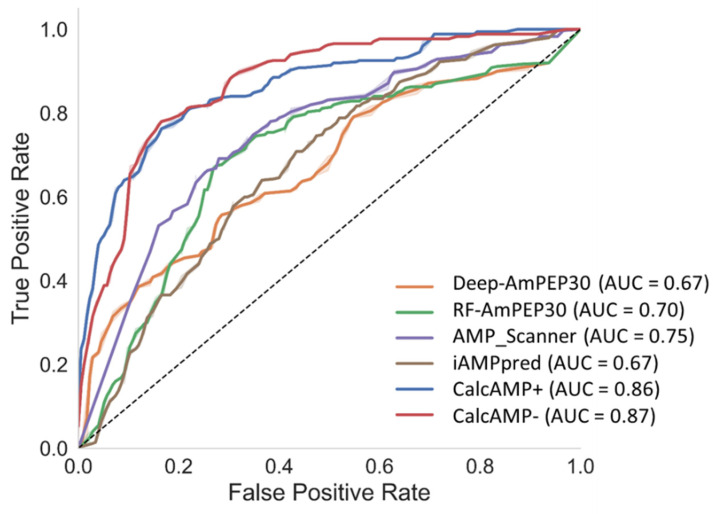
Receiver operator characteristic (ROC) curves of the different AMP classifiers and their area under the curve score.

**Figure 9 antibiotics-12-00725-f009:**
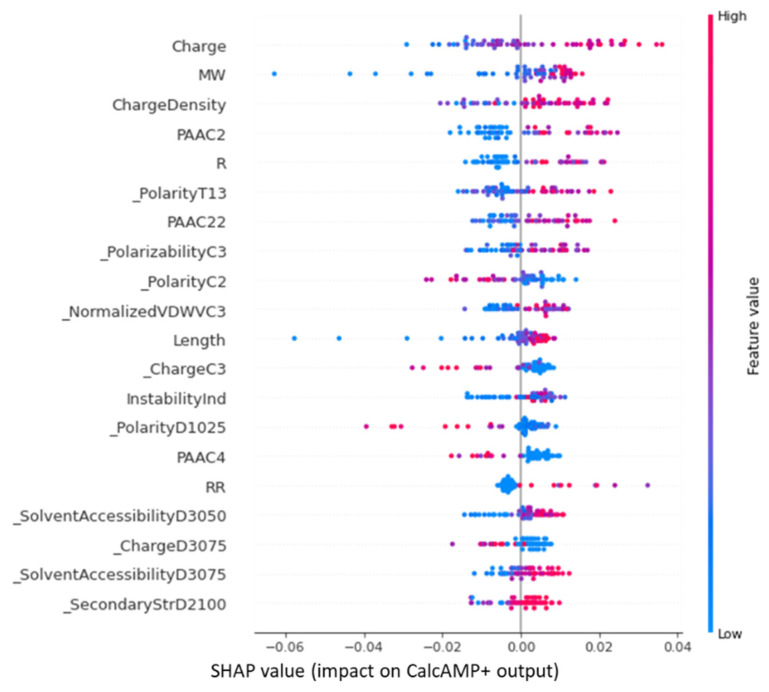
Top 20 feature importance plot and their impact on the external test set prediction for CalcAFP.

**Figure 11 antibiotics-12-00725-f011:**
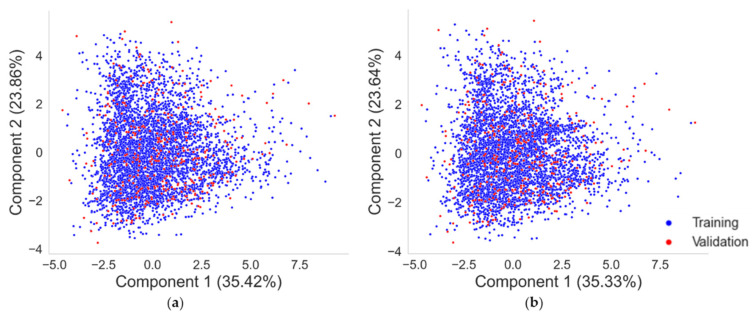
PCA projections of the training set (blue) and external test set (red) for Gram+ (**a**) and Gram− (**b**) categories.

**Table 1 antibiotics-12-00725-t001:** Comparison of different feature sets for Gram+ (white) and Gram− (grey) AMP prediction. The values in brackets represent the standard deviation obtained via 10-fold cross validation. Bold values indicate the best value per column.

Feature Set (#)	Accuracy	Sensitivity	Specificity	AUC-ROC	MCC
AAC (20)	0.77 (0.02)	0.77 (0.02)	0.78 (0.04)	0.85 (0.01)	0.55 (0.04)
	0.78 (0.02)	0.76 (0.01)	0.81 (0.03)	0.86 (0.02)	0.56 (0.03)
CTD (147)	0.79 (0.02)	0.77 (0.03)	0.82 (0.03)	0.87 (0.02)	0.59 (0.04)
	0.80 (0.01)	0.79 (0.02)	0.82 (0.02)	0.88 (0.01)	0.61 (0.03)
DPC (400)	0.77 (0.02)	0.78 (0.03)	0.76 (0.03)	0.85 (0.02)	0.53 (0.04)
	0.77 (0.02)	0.77 (0.02)	0.78 (0.03)	0.86 (0.01)	0.55 (0.04)
PseAAC (24)	0.77 (0.02)	0.76 (0.03)	0.79 (0.02)	0.85 (0.02)	0.55 (0.04)
	0.78 (0.02)	0.75 (0.03)	0.81 (0.02)	0.86 (0.01)	0.55 (0.04)
GPC (10)	0.78 (0.01)	0.78 (0.02)	0.79 (0.03)	0.85 (0.01)	0.57 (0.02)
	0.78 (0.01)	0.78 (0.03)	0.79 (0.02)	0.86 (0.01)	0.57 (0.02)
All (601)	**0.81 (0.02)**	**0.80 (0.03)**	**0.83 (0.03)**	**0.89 (0.02)**	**0.62 (0.05)**
	**0.81 (0.02)**	**0.80 (0.04)**	**0.82 (0.03)**	**0.89 (0.02)**	**0.62 (0.05)**

**Table 3 antibiotics-12-00725-t003:** Comparison of different AFP prediction classifiers using the external data set. Bold values indicate the best value per column.

Model	Accuracy	Sensitivity	Specificity	AUC-ROC	MCC
iAMPpred	0.52	**0.77**	0.27	0.56	0.04
ClassAMP	0.48	0.33	0.63	- ^1^	−0.03
Antifp	0.50	0.30	0.70	- ^1^	0.00
*Average*	*0.50*	*0.47*	*0.53*	*-*	*0.00*
CalcAFP	**0.77**	0.63	**0.90**	**0.86**	**0.55**

^1^ For ClassAMP and Antifp, AUC ROC could not be calculated since we do not have access to the probabilities associated.

**Table 4 antibiotics-12-00725-t004:** List of public AMP database used and their corresponding number of unique peptides.

Database	Number of Unique Sequence ^2^
ADAM 1 (A Database of Anti-Microbial Peptides) http://bioinformatics.cs.ntou.edu.tw/ADAM/index.html	7007
BaAMPs 1 (Biofilm-Active AMPs Database) http://www.baamps.it/	225
CAMP (Collection of Anti-Microbial Peptides) http://www.camp.bicnirrh.res.in/	8177
DBAASP (Database of Antimicrobial Activity and Structure of Peptides) https://dbaasp.org/	17,783
DRAMP (Data Repository of Antimicrobial Peptides) http://dramp.cpu-bioinfor.org/	22,259
LAMP2 (Linking Antimicrobial Peptides) http://biotechlab.fudan.edu.cn/database/lamp/index.php)	23,253
YADAMP (Yet Another Database of Antimicrobial Peptides) http://yadamp.unisa.it/	2525

^1^ In grey are the databases that were rejected. ^2^ accessed in March 2021.

## Data Availability

Data used in this study were uploaded in Zenodo: https://doi.org/10.5281/zenodo.7588702, code: https://github.com/CDDLeiden/CalcAMP.

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
