# Peer review of "CalcAMP: A New Machine Learning Model for the Accurate Prediction of Antimicrobial Activity of Peptides"

_antibiotics, 2023, doi:10.3390/antibiotics12040725_

Round 1
Reviewer 1 Report
The article describes the development of an antimicrobial prediction model using machine learning. The prediction model relies on the amino acid composition and physicochemical descriptors and algorithms with cross-validation and a holdout test dataset.
The authors used databases with experimental activity to classify as AMPs or Non-AMPs using an activity threshold. The peptide lengths were less than 35 residues.
The results are interesting and sound.
Minor points:
- The difference in molecular weight between AMPs and Non-AMPs:
How was the dispersion between the groups?
Was the difference statistically relevant?
- Figure 6 is not clear, and the legend does not include a description of the features.
Author Response
Response to Reviewer 1
“The article describes the development of an antimicrobial prediction model using machine learning. The prediction model relies on the amino acid composition and physicochemical descriptors and algorithms with cross-validation and a holdout test dataset.
The authors used databases with experimental activity to classify as AMPs or Non-AMPs using an activity threshold. The peptide lengths were less than 35 residues.
The results are interesting and sound.”
Minor points:
- The difference in molecular weight between AMPs and Non-AMPs:
How was the dispersion between the groups?
The standard deviation for both the molecular weight and the charge as a measure of dispersion between these two variables for AMPs and Non-AMPs was added to section 2.1.2: “As mentioned above, AMPs are significantly more positively charged than Non-AMPs (Mann-Whitney U test, p < 0.05). They have an average positive charge of +4.8 (standard deviation SD: 2.70) vs a charge of +2.8 (SD: 2.56) for Non-AMPs. AMPs are also significantly heavier (Mann-Whitney U test, p < 0.05) with a mean of 2241 g.mol-1 (SD: 703) vs 1911 g.mol-1 (SD: 715) for Non-AMPs, see figure 3c.” (p6, lines 182-187)
- Was the difference statistically relevant?
In order to be more accurate we performed a Mann-Whitney U for both the molecular weight and the charge differences between AMPs/Non-AMPs and reported the results in the manuscript as the differences were statistically relevant. See answer to the previous question (p6, lines 182-187).
- Figure 6 is not clear, and the legend does not include a description of the features.
Figure 6 displays a standard SHAP analysis output generated using the original Python library. We have reviewed the figure and we are not convinced that modifications are necessary since we also do not wish to alter the original output further. However, to enhance clarity, we have revised the text to provide a clearer description, especially for the CTD descriptors legend. Therefore, the following sentence has been added to the text (and the figure legend): “The majority of the top features are from CTD descriptors. They are identifiable by their names beginning with an underscore character, followed by the property and finally the component characteristics: composition (C), transition (T) and distribution (D).” (p9, lines 314-317)
Reviewer 2 Report
The paper structure and motivation are fine, and the theme is relevant. The conclusions are supported by the results and the discussion is also supported by literature. However, I advise the authors to improve the several sentences, specially on the introduction section.
In my opinion the paper should be accepted after minor revisions.
Minor
Line 17: Rephrase the sentence. For example: Among them are defence peptides that can be (…)
Line 45 to 47: The 3 sentences starts with “they”. Please improve them.
Line 89/90 : Please improve the sentences structures.
Line 101/103: First and second instead of Firstly and secondly. Improve the sentence after secondly. Who are their?
Line475: Specify what do you mean with “more”.
Author Response
Response to Reviewer 2
“The paper structure and motivation are fine, and the theme is relevant. The conclusions are supported by the results and the discussion is also supported by literature. However, I advise the authors to improve the several sentences, specially on the introduction section.
In my opinion the paper should be accepted after minor revisions.”
Minor points:
- Line 17: Rephrase the sentence. For example: Among them are defence peptides that can be (…)
The sentence was rephrased as proposed: “Among them are defense peptides with the ability to target a wide range of pathogenic organisms, including bacteria, viruses, parasites, and fungi.” (p1, lines 17-18)
- Line 45 to 47: The 3 sentences starts with “they”. Please improve them.
The sentences were improved and rephrased so that they do no start with “they”: “These peptides, which are already present in the innate immune system of plants, animals, and humans, possess both antimicrobial activity and immunomodulatory properties [7]. AMPs are an essential component of the body's first line of defense against pathogens, even before the adaptive immune system is activated. Moreover, they exhibit diverse structural and functional profiles that can be optimized and fine-tuned to enhance their activity further [8,9]. As a result, they offer tremendous potential as novel therapeutic agents for combating a wide range of pathogens. (p1-2, lines 45-56).
- Line 89/90 : Please improve the sentences structures.
The paragraph has been rewritten (p3, lines 95-114).
- Line 101/103: First and second instead of Firstly and secondly. Improve the sentence after secondly. Who are their?
The changes were made accordingly and the text of the paragraph improved:.: “First, they do not account for differences in bacterial species or membrane structure differences, whereas the potency of AMPs can vary significantly depending on the target and the type of bacterial envelope [35]. Second, these tools employ randomly selected sequences without experimentally determined antimicrobial activity as negative dataset, rather than using confirmed inactive peptides.” (p3, lines 116-121)
- Line475: Specify what do you mean with “more”.
More was used in this context as “etc”. since there are much more factors that can change the outcome of experimental tests. It was deleted for more clarity, and the sentence is now as follows: “Indeed, several factors can influence the outcome, such as the bacterial strain tested, the growth medium, the type of activity measured.” (p15, lines 499-501